Trends and challenges in organoid modeling and expansion with pluripotent stem cells and somatic tissue

Ge Jian-Yun 1 2 3
Wang Yun 4 5
Li Qi-Lin 6
Liu Fan-Kai 7
Lei Quan-Kai 4
Zheng Yun-Wen zhengyunwen@ihcams.ac.cn 1 2 4 8 9
1 Guangdong Provincial Key Laboratory of Large Animal Models for Biomedicine, and South China Institute of Large Animal Models for Biomedicine, School of Pharmacy and Food Engineering, Wuyi University , Jiangmen , Guangdong , China
2 Haihe Laboratory of Cell Ecosystem, Institute of Hematology, Chinese Academy of Medical Sciences , Tianjin , China
3 Innovation and Transformation Center, University of Traditional Chinese Medicine , Fuzhou , Fujian , China
4 Institute of Regenerative Medicine, and Department of Dermatology, Affilated Hospital of Jiangsu University , Zhenjiang , Jiangsu , China
5 Department of Dermatology, The First People’s Hospital of Changzhou , Changzhou , Jiangsu , China
6 State Key Laboratory of Bioreactor Engineering, East China University of Science and Technology , Shanghai , China
7 Institute of Translational Medicine, Medical College, Yangzhou University , Yangzhou , Jiangsu , China
8 Department of Medicinal and Life Sciences, Faculty of Pharmaceutical Sciences, Tokyo University of Science , Noda , Chiba , Japan
9 Division of Regenerative Medicine, Center for Stem Cell Biology and Regenerative Medicine, Institute of Medical Science, The University of Tokyo , Tokyo , Japan
Zhang Xin
Electronic publication date: 2024 Nov 27
Publication date: 2024
Volume: 12
Electronic Location ID: e18422
Received 2024 Apr 28; Accepted 2024 Oct 8
Copyright: ©2024 Ge et al.
Copyright year: 2024
Copyright holder: Ge et al.
License: This is an open access article distributed under the terms of the Creative Commons Attribution License, which permits unrestricted use, distribution, reproduction and adaptation in any medium and for any purpose provided that it is properly attributed. For attribution, the original author(s), title, publication source (PeerJ) and either DOI or URL of the article must be cited.
License URL: https://creativecommons.org/licenses/by/4.0/

Keywords: Organoid, Up-scaling, Expansion, Disease modeling, Transplantation, Bioreactor, Gene-editing, Universal biobanking, Induced pluripotent stem cells, Somatic stem cells

Funding: National Natural Science Foundation of China 82270697 82070638 82370517 Guangdong Basic Applied Basic Research Foundation 2023A1515012574 Jiangsu Provincial Medical Key Discipline Cultivation Unit JSDW202229 Science and Technology Planning Project of Guangdong Province of China 2021B1212040016 Grant for International Joint Research Project of the Institute of Medical Science, University of Tokyo All the external funding or sources of support received during this study: National Natural Science Foundation of China (82270697, 82070638 and 82370517) and Guangdong Basic Applied Basic Research Foundation (2023A1515012574), Jiangsu Provincial Medical Key Discipline Cultivation Unit (JSDW202229), the Science and Technology Planning Project of Guangdong Province of China (2021B1212040016), and the Grant for International Joint Research Project of the Institute of Medical Science, University of Tokyo. No other external funding was received for this study. The funders had no role in study design, data collection and analysis, decision to publish, or preparation of the manuscript.

==============================
The increasing demand for disease modeling, preclinical drug testing, and long waiting lists for alternative organ substitutes has posed significant challenges to current limitations in organoid technology. Consequently, organoid technology has emerged as a cutting-edge tool capable of accurately recapitulating the complexity of actual organs in physiology and functionality. To bridge the gaps between basic research and pharmaceutical as well as clinical applications, efforts have been made to develop organoids from tissue-derived stem cells or pluripotent stem cells. These developments include optimizing starting cells, refining culture systems, and introducing genetic modifications. With the rapid development of organoid technology, organoid composition has evolved from single-cell to multi-cell types, enhancing their level of biomimicry. Tissue structure has become more refined, and core challenges like vascularization are being addressed actively. These improvements are expected to pave the way for the construction of organoid atlases, automated large-scale cultivation, and universally compatible organoid biobanks. However, major obstacles remain to be overcome before urgently proof-of-concept organoids can be readily converted to practical applications. These obstacles include achieving structural and functional summarily to native tissue, remodeling the microenvironment, and scaling up production. This review aims to summarize the status of organoid development and applications, highlight recent progress, acknowledge existing limitations and challenges, and provide insights into future advancements. It is expected that this will contribute to the establishment of a reliable, scalable, and practical platform for organoid production and translation, further promoting their use in the pharmaceutical industry and regenerative medicine.

Introduction

With recent technological advances, organoids have emerged as the leading in vitro model for the detailed exploration of the molecular landscape of human organs. This advancement enables in-depth and systematic dissection of human biological systems and disease mechanisms, accelerating drug discovery and the development of therapeutic strategies. Organoids are in vitro organ substitutes generated exclusively from either somatic stem cells (SSC) or pluripotent stem cells (PSCs) through self-organization, which forms a complex three-dimensional (3D) architecture with physiological features that closely resemble native organs. These features make organoid models more accessible and practical compared to animal models, providing deeper insights into organ biology and the progression of related disease development (Rossi, Manfrin & Lutolf, 2018).

Organoids derived from SSCs are biocompatible with the host tissue or organ, making them convenient sources, particularly for autologous transplantation. However, SSCs present practical challenges, including difficulties in separation and purification, as well as limited capacity for in vitro expansion and lineage differentiation. As an alternative, PSC-derived induced pluripotent stem cells (iPSCs) offer significant potential for broad applications due to their theoretically permanent replication power (Cao et al., 2021). More importantly, their unique multilineage differentiation potential allows the generation of multiple organ-forming cell components, enabling the reconstruction of vascularized multicellular organoids with neuroendocrine (Lamers et al., 2021) and immune systems (Xu et al., 2018) under specific differentiation and co-culture protocols. These organoids can mimic the physiological compositions and functions of native organs, expanding their potential for pharmacological and therapeutic application (Takebe et al., 2017). However, preparing individual cell types remains time-consuming, labor-intensive, and impractical for widespread use. Recently, gene editing has enabled a single initial PSC population to differentiate into multiple cell composition under pre-designed gene network regulation, ultimately forming engineered tissue with expansion capabilities. Although still in its early stages, this breakthrough offers important insights for constructing a more efficient, labor-reduced, and scalable organoid production system.

In the past 20 years, emerging evidence has shown that organ-like substitutes can recapitulate human pathologies, offering breakthroughs in disease modeling and drug development (Rossi, Manfrin & Lutolf, 2018). Several pioneering studies have demonstrated that patient-derived organoids could mimic the genetic and pathological features of the original diseased tissue (Dekkers et al., 2013; Dutta, Heo & Clevers, 2017; Huch et al., 2015; Lancaster et al., 2013; Nie et al., 2018a; Ouchi et al., 2019; Zhou et al., 2017). More recently, gene modification techniques, such as CRISPR/Cas9, have enabled the use of organoids from healthy donors to establish disease models including cancer (Matano et al., 2015; Sun et al., 2019). Proof-of-concept studies also suggest that organoids can serve as alternative grafts for regenerative medicine, helping to rescue or recovery damaged issues (Nie et al., 2018b; Shirai et al., 2016), offering new hope for treating previously incurable diseases. However, variations in cell sources and protocols between research groups lead to differences in organoid structure and function, affecting the accuracy and reproducibility of these reported disease models. This inconsistency is a major challenge for rapidly growing organoid industry (Rossi, Manfrin & Lutolf, 2018).

In recent years, expectations for the practical applications of organoids have grown steadily. However, this fulfillment relies greatly on critical breakthroughs in the reliability, reproducibility, and practicability of organoid generation. Therefore, the key challenges remain in bridging the gap between in vitro organoids and native organs, as well as translating laboratory studies into practical production and applications. This review summarizes recent progress in organoid generation, focusing on disease modeling and transplantation therapies. Simultaneously, we discuss challenges related to addressing architectural and functional complexity, enhancing maturation and establishing reproducible and scalable culture systems. Notably, we highlight advances in organoid atlases, automated cultures, and universally compatible iPSC-organoid biobanks, which could greatly improve drug development and personalized treatments in the future.

Survey Methodology

We searched PubMed for articles on trends and challenges in organoid modeling and expansion with pluripotent stem cells and somatic tissue. Keywords included “Organoid, up-scaling, expansion, disease modeling, transplantation, bioreactor, gene-editing, universal biobanking, induced pluripotent stem cells, somatic stem cells, three-dimension, multicellular organization, extracellular matrix” yielding over 8,000 results. The articles were screened individually to assess relevance to our research. After organizing and analyzing the results, we downloaded 240 literature reviews. Of these, 178 articles were cited in this review, while 62 articles were excluded as irrelevant. Most of the literature cited was published in the past decade. This review aims to serve as a reference for researchers working on organoid research and developing tools for drug screening and transplantation therapy using iPSC-derived multicellular organoids.

Current achievements

Here we summarize the current state of research on organoid production, the impact of co-culture systems, the in vitro microenvironment on organoid production, and current research hotspots in organoid culture. Additionally, we reviewed the current applications of organoids in disease modeling and their potential as disease therapeutic tools.

Organoid microenvironment simulation (multicellular organization and extracellular matrix)

Co-culturing isolated tissue stem cells or dissected tissue fragments is the principal strategy for generating SSC-derived organoids. Without additional modifications, such as gene editing, these organoids are relatively easy to manage and maintain, showing strong potential for mimicking adult tissue biology. Among pioneering studies, a single LGR5+ stem cell population was shown to reconstitute mouse and human intestines or liver organoids when cultured with defined niche factors (mainly growth factors and Matrigel) (Huch et al., 2015; Sato et al., 2009). Using single-cell sequencing, a novel protein C receptor-positive (Procr+) cell population was identified from an adult mouse pancreas, which robustly formed islet-like organoids when cultured at clonal density (Wang et al., 2020). Notably, these SSC-derived organoids could be expanded in a defined culture system without genome instability and could be differentiated into functional cells in vitro or upon transplantation in vivo (Huch et al., 2015; Wang et al., 2020). However, identifying SSC markers and isolating and purifying SSCs are often challenging and labor-intensive. In addition, SSCs exclusively produce organoids with a single epithelial population, lacking the complex multilineage compositions found in real organs, such as the vascular, immune, and nervous systems (Calandrini & Drost, 2021; Mohammadi et al., 2021). Although technically challenging, PSC-derived organoids are becoming more accessible and practical. With proper differentiation protocols, PSCs could develop into various desired cell types within specific tissues. Notably, complex multilineage organoid systems could be developed from a single PSC source, capturing cellular heterogeneity and interactions necessary to replicate the pathology of human organs (Silva et al., 2019). To date, multiple organoid types have been generated from PSCs (Table 1), demonstrating their wide-ranging applications.

Table 1 Summary for PSC or tissue derived organoids and their application.

Organoid types	Initial cells	3D culture: embedded in gel or suspended in a plate	Applications & Significance	Author(s),year	
Liver	PHHs & liver NPCs	BME	Hepatotoxicity test	Messner et al., 2013	
	hiPSCs	Matrigel	Generation of a vascularized and functional human liver bud from PSCs	Takebe et al., 2013	
	Patient liver cells	BME	Modelling of primary liver cancer	Broutier et al., 2017	
	hiPSCs	ULA plate	Modelling of liver development	Takebe et al., 2017	
	PMHs	ULA plate	Recapitulation of liver regeneration potential	Peng et al., 2018	
	PMH & PHH	Matrigel	Recapitulation of the proliferative damage-response of hepatocytes	Zilch et al., 2018	
	hiPSCs	ULA plate	Modelling of hepatitis B virus infection	Nie et al., 2018a	
	hPSCs	Matrigel	Prediction of toxicity and the evaluation of drugs for hepatic steatosis	Mun et al., 2019	
	Patient liver cells	BME	Drug screening for anti-HBV activity and drug-induced toxicity	De Crignis et al., 2021	
	PHHs	Matrigel	High-throughput screening of chemical and food-derived compounds with anti-hyperuricemic bioactivity	Hou et al., 2022	
	hiPSCs	Matrigel	High-throughput	Shrestha et al., 2024	
Pancreas	Mouse pancreatic epithelial cells	Matrigel	Modelling of pancreatic development	Huch et al., 2013	
	Patient tumor cells	Matrigel	Modelling of pancreatic tumorigenesis	Boj et al., 2015	
	Mouse islet EpCAM+ cells	Matrigel	Rescuing streptozotocin (STZ)-induced diabetes in mice	Wang et al., 2020	
	hiPSCs	Matrigel	Rescuing streptozotocin (STZ)-induced diabetes in mice	Yoshihara et al., 2020	
	hPSCs	Matrigel	Modelling of the development of exocrine pancreas	Huang et al., 2021	
	hPSCs	Matrigel	Modelling of the development of hepato-biliary-pancreatic	Koike et al., 2019	
Lung	hPSCs	Matrigel	Modelling of lung development	Dye et al., 2015	
	hPSCs	Matrigel	Modelling of lung development	Miller et al., 2018	
	hPSCs	Matrigel	Modelling of fibrotic lung disease	Strikoudis et al., 2019	
	Patient pulmonary cells	Matrigel	Modelling of non-small cell lung cancer	Shi et al., 2020	
	Patient pulmonary cells	Matrigel	Prediction of the targeted and the chemotherapeutic drugs	Hu et al., 2021	
	hPSCs	ULA plate & Matrigel	Modelling of SARS-CoV-2 infection on lung	Han et al., 2021	
	hPSCs	Matrigel	Modeling fibrotic alveolar transitional cells	Ptasinski et al., 2023	
Intestine	mASCs	Matrigel	Building crypt-villus structures in vitro without a mesenchymal niche	Sato et al., 2009	
	hiPSCs	Matrigel	Modelling of human intestine development	Spence et al., 2011	
	hiPSCs	Matrigel	Modelling of congenital loss of intestinal enter-oe ndocrine cells	Fordham et al., 2013	
	hPSCs	Hydrogel	Repairing of intestinal injury	Cruz-Acuña et al., 2017	
	Patient intestinal cells	Matrigel	Testing of anti-inflammatory drugs	d’Aldebert et al., 2020	
	Patient intestinal cells	Matrigel	Drug screening for intestinal diseases	Cho et al., 2021	
	hPSCs	ULA plate & Matrigel	Modelling of SARS-CoV-2 infection on colon	Han et al., 2021	
	Patient intestinal cells	Matrigel	Modelling of Cronkhite-Canada Syndrome	Poplaski et al., 2023	
Brain	mESCs	ULA plate	Modelling of the development of polarized cortical tissue	Eiraku et al., 2008	
	hPSCs	ULA plate	Modeling of microcephaly	Lancaster et al., 2013	
	hPSCs	ULA plate	Modelling of neural development and disease progressing	Birey et al., 2017	
	Patient glioblastoma cells	ULA plate	Modelling of glioblastomas	Jacob et al., 2020b	
	hPSCs	ULA plate	Modelling of SARS-CoV-2 infection on brain	Jacob et al., 2020a	
	hiPSCs	Matrigel	Modelling of brain development	Gabriel et al., 2021	
	hiPSCs	Matrigel in a perfusion plate	Assessment of developmental neurotoxicity	Acharya et al., 2024	
	hiPSCs	Matrigel	Modeling of HIV-1 infection and NeuroHIV	Donadoni et al., 2024	
Retina	hESCs	ULA plate & Matrigel	Modelling of optic cups and layered stratified neural retina development	Nakano et al., 2012	
	hiPSCs	Geltrex	Modelling of retinal development	Xie et al., 2020	
	hESCs	ULA plate	Modelling of retinal development	Savoj et al., 2022	
	hiPSCs	ULA plate	Modelling of Alzheimer’s disease neuropathology	James et al., 2024	
Skin	mPSCs	ULA plate	Modelling of skin diseases and revealing of hair follicle induction, hair growth	Lee et al., 2018	
	hPSCs	ULA plate	Modelling of the cellular dynamics of developing human skin	Lee et al., 2020a	
	hiPSCs	Matrigel	Testing of skin-related drugs	Ebner-Peking et al., 2021	
	Mouse epidermal cells	ULA plate	Modelling of self-organization into tissue patterns of stem cells in organoids	Lei et al., 2023	
	hiPSCs	ULA plate	Modelling of human skin development, disease and reconstructive surgeries	Shafiee et al., 2023	
Kidney	Embryonic kidney cells	Cell pellet	Modelling of organotypic renal structures by self-organization	Unbekandt & Davies, 2010	
	mESCs & hiPSCs	ULA plate	Modelling of kidney organogenesis	Taguchi et al., 2014	
	hiPSCs	Matrigel	Screening of nephrotoxicity	Takasato et al., 2015	
	Human kidney tubular epithelial cells	Matrigel & BME	Modelling of infectious, malignant and hereditary kidney diseases	Schutgens et al., 2019	
	Patient renal cells	Matrigel	Modelling of renal cancer	Grassi et al., 2019	
	hESCs	ULA plate	Modelling of flow-enhanced vascularization	Homan et al., 2019	
	Mouse ureteric bud progenitors	Matrigel	Modelling of congenital anomalies of kidney and urinary tract	Zeng et al., 2021	
	hiPSCs	Matrigel	Modeling of Fabry disease nephropathy	Cui et al., 2023	
	hiPSCs	Matrigel	Modeling of FAN1-deficient kidney disease	Lim et al., 2023	
Bone	hPDCs	Agarose microwell	Rescue tibia defects of mice	Nilsson Hall et al., 2020	
	hPSCs	Matrigel	Bone healing	Tam et al., 2021	
	PHOs	Matrigel	Modelling of skeletal development	Abraham et al., 2022	
	Human cartilage and bone tissues	Matrigel	Modeling of tissue development and disease	Abraham et al., 2022	
	BMSCs	Hydrogel	Rapid bone defect regeneration and recovery	Xie et al., 2022	
Heart	hPSCs	Matrigel	Two pro-proliferative small molecules without detrimental impacts	Mills et al., 2019	
	mESCs	Matrigel	Modelling of carcinogenesis	Lee et al., 2020b	
	hESCs	ULA plate & Matrigel	Modelling of the development of early heart and foregut	Drakhlis et al., 2021	
	hPSCs	ULA plate	Modeling of cardiac development and congenital heart disease	Lewis-Israeli et al., 2021	
	hiPSCs	ULA dish & Matrigel	Recapitulate morphological/functional aspects of the heart	Lee et al., 2022	
	hPSCs	ULA plate	Modeling of syntheticheart development and cardiac disease	Volmert et al., 2023	
Notes.

BME basement membrane extract

D dimensional

ESC embryonic stem cell

h human

iPSC induced pluripotent stem cell

m mouse

NA not available

PHCs primary human cholangiocytes

NPCs non-parenchymal cells

PHHs primary human hepatocytes

PHOs primary human osteocytes

PMHs primary mouse hepatocytes

PSCs pluripotent stem cells (iPSC & ESC)

ASCs adult stem cells

ULA ultra-low attachment

Single-cell population organoids do not fully replicate the complexity of real organs because they lack native stromal cells and blood vessels, which are essential for tissue development and maturation (Yu, 2020). To recapitulate the multicellular interactions in the liver, initial attempts used human mesenchymal stem cells (MSCs) and umbilical vein endothelial cells (HUVECs) to substitute the liver’s intrinsic stromal components, such as hepatic stellate cells, and endothelial cells. This approach successfully established vascularized multicellular liver organoids through self-cell sorting and architectural rearrangements, and these organoids exhibited enhanced liver function due to their multicellular system-autonomous endogenous signaling (Takebe et al., 2013). Subsequently, to avoid allogeneic cell source integration, the generation of liver parenchymal and major supportive cell lineages was induced from a single PSC. Interestingly, this all-iPSC-based multicellular organoid demonstrated a higher level of structural and functional similarity to primary liver tissue compared to the former model, suggesting that self-assembly with an autologous cell population may promote maturation in general, although the underlying mechanism remains to be clarified (Takebe et al., 2017).

However, organoids created by co-culturing separate pre-established cell types are considered structurally irregular and disordered compared to native organs, where multilineages are developed simultaneously. Moreover, the asynchronously developed aggregation may lack specific intercellular signal interactions found in tissues, which are essential for fully recapitulating biological functions. Therefore, multicellular co-development was recently achieved when gene editing was adopted. By precisely regulating gene networks through PROX1 and ATF5 overexpression and CYP3A4 activation, a single iPSC population can now develop into both liver parenchymal and nonparenchymal cell populations at the same time during cell aggregation. Importantly, these liver organoids were well vascularized, exhibited mature hepatic functions, and responded effectively to perturbations and feedback regulation (Velazquez et al., 2021). However, current multicellular organoid systems still face challenges. Notably, they lack immune cells and exhibit differences in cell distribution and ratios compared to natural organs, which vary depending on the protocols used in different laboratories (Ouchi et al., 2019; Velazquez et al., 2021). To improve organoid models, a deeper understanding of the multilineage developmental trajectories in a specific organ and the development of advanced gene editing tools are needed. This will help build precise cell types in organoids with faithful ratios, spatial distributions, and organization patterns.

Among in vitro culture environments, the extracellular matrix (ECM) may be the most crucial and adjustable factor that influences organoid features such as spatial architecture, growth, maturation, and even carcinogenesis. Matrigel is currently the most used natural ECM and is known for promoting organoid constitution and growth. However, its undefined composition and potential risks from immunogen and pathogen limit its clinical applications (Rossi, Manfrin & Lutolf, 2018). The development of designer matrices, particularly synthetic hydrogels with fine-tunable and controlled biophysical properties, has rapidly advanced in recent years. For instance, modular synthetic polyethylene glycol (PEG) hydrogels have been designed for culturing intestinal organoids. Notably, regulating the stiffness of the matrix backbone, known as mechanically dynamic system, researchers can influence the expansion, differentiation, and organogenesis of these organoids (Gjorevski et al., 2016). Subsequently, a composition-defined PEG-based hydrogel system was designed based on functional analysis of cellular adhesion in pancreatic cancer cells. This system revealed the functional role of critical ECM signaling in supporting both normal and cancerous pancreatic organoid growth (Below et al., 2022). Despite these efforts, identifying all the relevant ECM signaling molecules require for mimicking native tissue developmental trajectory or disease progression remains a major challenge. For this purpose, naturally derived matrices from decellularized tissues are being explored. An ECM hydrogel derived from decellularized porcine small intestine mucosa/submucosa was found to support the formation and growth of a wide range of endoderm-lineage organoids (Giobbe et al., 2019). In addition, several studies have identified the supportive roles of ECM from various tissues, including the liver (Zahmatkesh et al., 2021), pancreas (Bi et al., 2020), kidneys (Garreta et al., 2024) and brain (Simsa et al., 2021) in promoting the formation and maturation of PSC-organoids in specific germ layers. While the tissue ECM provides the most supportive and comprehensive signaling for organoid culture, closely mimicking the native microenvironment, the undefined compositions of these tissues necessitate thorough identification and rigorous biosafety verification before they can be used in clinical application.

Notably, the native organ microenvironment plays a crucial role in regulating the dynamic development of tissue homeostasis, regeneration, and pathogenesis, with each organ requiring a specific microenvironment. To address this, researchers have turned to microfluidic culture systems because of their advantages in miniaturization, integration, and low reagent consumption. These systems not only model the dynamic 3D in vivo microenvironment but also enable precise control of multiple parameters of the variables in the environment, such as the concentration gradient of the fluid shear stress. This allows for the creation of a controllable physiological environment for specific organoid development and maintenance (Jalili-Firoozinezhad, Miranda & Cabral, 2021). However, most established microfluidic culture systems cannot satisfy all the elements that make up a complex in vivo microenvironment. For example, while some reports have revealed the unique cellular organization, ECM composition, and signaling in organoids at specific growth or differentiation stages, aspects such as blood flow, oxygen, and metabolic exchange are often overlooked in most culture system designs. Combining the utility of 3D scaffolds, micropatterning, and advanced culture technologies such as organ-on-chip systems to more accurately mimic in vivo dynamics remains a significant challenge (Zhang et al., 2017). With optimized parameter design and intelligent controls, an integrated culture system would undoubtedly enhance the resemblance between organoids and native organs.

Applications of organoids (disease modeling and transplantation therapy)

Undoubtedly, ongoing research efforts to promote the structural and functional similarity between organoids and natural organs are aimed at realizing their vast potential for biomedical applications, particularly in drug testing and therapy. To some extent, in vitro models could recapitulate important aspects of human organs in physiology and pathology, providing a valuable tool for studying organ development, pathogenesis, and drug responses, areas that are difficult or impossible to explore directly in human subjects (Bock et al., 2021).

Challenges in cancer research and drug development is mainly due to their heterogeneity, including diverse mutational, epigenetic, and metabolic profiles, as well as the complexity of the tumor environment (McGranahan & Swanton, 2017). Tumor organoids, directly derived from patient-resected tumors and biopsies, offer a novel approach to capturing the original characteristics of tumors. They are considered more cost-effective and efficient compared to traditional animal models or patient-derived tumor xenografts. Pancreatic cancer is one of the most lethal malignancies owing to its late diagnosis and limited treatment option. To better understand the developmental cues of pancreatic tumorigenesis, patient-derived pancreatic organoids have been established as a tractable system to identify molecular and cellular characteristics at various stages of the disease. Both in vitro and transplantation experiments have confirmed a similar cluster of genes and pathways altered during disease progression, suggesting the faithfulness of this organoid model (Boj et al., 2015). In another study, a more detailed comparison between tumor and tumor-derived organoids were identified, including differentiation status, histoarchitecture, phenotypic heterogeneity, and patient-specific physiological changes. Notably, the observed correlation between tumors and their matched organoids in terms of sensitivity to histone methyltransferase EZH2 inhibitors highlights the potential of organoids for personalized drug screening and precise therapy. Moreover, the same group demonstrated these results using PSC-derived pancreatic tumors with mutations in KRAS and TP53 (Sun et al., 2019). Another group established human CRC tumor-derived organoids that well represent both morphological and molecular heterogeneities of original tumors. A robust organoid-based drug screening system was developed to efficiently identify repurposed drugs for CRC (Mao et al., 2024).Similarly, liver organoids generated from patients with Alagille syndrome were used to model in vivo pathology (Huch et al., 2015). Collectively, PSCs combined with gene editing largely broadens the organoid source, not limiting it to patient donors, and are particularly practical for establishing cancer models with defined mutation signatures.

Organoids have been extensively used for hereditary and infectious disease modeling. For instance, intestinal organoids generated from patients with cystic fibrosis, a disease caused by mutations in the cystic fibrosis transmembrane conductance regulator (CFTR), were used for drug screening for individual patients (Dekkers et al., 2013). Similarly, liver organoids generated from patients with alpha-1-antitrypsin deficiency and Alagille syndrome have been used to model in vivo pathology (Huch et al., 2015). Extensive research has been conducted on these pathogens since the emergence of a new coronary pneumonia ailment. Lung organoids and brain organoids (Cakir et al., 2019; Velasco et al., 2019) have proven to be effective tools for studying SARS-CoV-2-related pathogenesis and treatment strategies. Brain organoids are also used to model microcephaly (Lancaster et al., 2013) and Zika infection (Zhou et al., 2017). Kidney organoids have significantly enhanced the capability to discover novel disease mechanisms and validate candidate drugs for clinical translation in polycystic kidney disease (PKD) (Liu et al., 2024). Many research groups, including ours, have explored the potential of hPSC-derived liver organoids to model hepatitis B virus (HBV) infection and steatohepatitis (Nie et al., 2018a; Ouchi et al., 2019). Our findings highlight the importance of using multicellular organoids to increase susceptibility to HBV infection, which is believed to be the key step in recapitulating the virus life cycle (Cao et al., 2021). Nevertheless, additional cellular components, such as immune cells, are required to create a more accurate modeling system. Another promising application of organoids is in the study of rare diseases, where CRISPR-based gene editing and patient-derived iPSCs offer a personalized approach to disease modeling, such as alpha-1-antitrypsin deficiency and Rett syndrome (Gomes et al., 2020; Gómez-Mariano et al., 2020). These models allow researchers to explore rare genetic conditions at the molecular level, often revealing new therapeutic strategies for diseases that were previously under-researched.

Organoids play a crucial role in drug metabolism and toxicology research by offering human-relevant models that simulate the absorption, distribution, metabolism, and excretion of drugs (Wang et al., 2021a). Liver organoids, which exhibit key drug-metabolizing enzymes such as cytochrome P450, provide a more accurate platform for predicting drug behavior in preclinical trials, improving the drug development pipeline. Additionally, liver and kidney organoids are increasingly used in toxicology studies to predict organ-specific toxicities. This approach reduces reliance on animal models and enhancing the precision of safety evaluations (Czerniecki et al., 2018; Zhang et al., 2023).

Functional organoids hold promise as alternative substitutes for transplantation in therapeutic strategies. Our group is among the first to perform the transplantation of multicellular PSC-derived liver organoids to treat liver damage in a mouse model. We confirmed the integration and maturation of these vascularized organoids within the host, and demonstrated that they could improve survival and liver function in mice with acute liver failure (ALF), providing a proof-of-concept regimen for treating severe or late-stage liver diseases (Nie et al., 2018b; Nie et al., 2018a). Importantly, our findings of in vivo environment-conducted organoid maturation were consistent with the results of another transplantation trial, in which cholangiocyte organoids displayed transcriptional diversity from primary human cholangiocytes after culture; furthermore, they could regain their in vivo signatures when transplanted back in their physiological position (Sampaziotis et al., 2021). Similarly, PSC-derived kidney organoids were found to induce neovascularization and significant maturation of glomeruli and tubules after subrenal transplantation (Hickey et al., 2019). ESCs- derived retinal organoids have been used to transplant organoid-derived RGCs into the murine eyes, achieving long-distance regeneration and functional connectivity remains a challenge (Rao et al., 2025). Despite the considerable potential of regenerative medicine, generating a sufficient number of high-quality organoids remains a significant challenge for most research groups.

Difficulties and challenges

Scaled culture system

Over the years, organoid culture systems have proven to be a revolutionary paradigm for drug development and regenerative medicine, but they have not yet been widely applied in the pharmaceutical and industrial fields. One major obstacle is the scalability and reproducibility. To amplify batch production, a microwell-array culture platform was initially proposed for massive organoid production in a single culture, with over 20,000 micro spots in a single well with an optimized size and space design, which would theoretically support a clinically relevant batch scale (>108 cells/batch). However, this production system did not resolve the substantial issues associated with scaling organoid quantities. Moreover, the labor-intensive and inefficient manual operation for iPSC differentiation and the co-culturing of multiple cell populations was barely feasible and reproducible for practical batch-to-batch applications (Cao et al., 2021).

Expanding the number of organoids in scaled culture systems is considered a more practical and effective strategy for organoid production. As listed in Table 2, various pioneering studies have shown that organoids derived from certain tissue-derived stem and fetal cells (especially from the liver) have long-term expansion capabilities in vitro, although their lifespan is often limited. As a promising source, PSC-derived organoids have demonstrated a more powerful expansion capability, with increasing achievements, including those in the endodermal layer (Akbari et al., 2019; Giobbe et al., 2019; Mun et al., 2019; Yamamoto et al., 2017). However, these seemingly promising results have common drawbacks. Organoids are exclusively cultured in Matrigel or tissue-derived hydrogel layers. This method hampers organoid manipulation and testing, as well as clinically relevant applications such as transplantation and post-tissue engineering. Additionally, the static culture system restricts timely material and signaling transport between organoids and the culture environment.

Table 2 Up-to-date strategies for organoid expansion of tissue derived cells from the three germ layers.

Organoid types	Initial cells	3D Expansion systems: Embedded in a gel or suspension in a plate	Expansion capability: Passage days/expansion duration	Medium and supplements	Author(s),year	
Liver	hiPSC-Heps, HUVECs, hMSCs	Matrigel	4–6 d/NA	EGM, HCM, dexamethasone, OSM, HGF	Takebe et al., 2013	
	PMHs & PHHs	Matrigel	7–10 d/>6 mo	AdDMEM/F12 medium, B27, N-Acetylcysteine, gastrin, RSPO1, Noggin, Wnt, EGF, FGF7, FGF10, HGF, TGFa, Nicotinamide, A83-01, CHIR99021, Y27632	Hu et al., 2018	
	PHHs	BME gel	7-10d/9–12 mo	AdDMEM/F12 medium, B27, N-Acetylcysteine, gastrin, RSPO1, Noggin, Wnt, EGF, FGF10, FGF19, HGF, Nicotinamide, A83-01, FSK, Y27632	Artegiani et al., 2019	
	hiPSC-derived EpCAM+hepatic progenitors	Matrigel	7d/3 mo	AdDMEM/F12 medium, B27, N-Acetylcysteine, gastrin, RSPO1, Noggin, Wnt, EGF, FGF10, HGF, Nicotinamide, A83-01, FSK, Y27632	Akbari et al., 2019	
	hPSCs	Matrigel	10d/17 d	AdDMEM/F12 medium, N2, B27, N-Acetylcysteine, gastrin, RSPO1, EGF, FGF10, HGF, Nicotinamide, A83-01, FSK	Mun et al., 2019	
	hPSCs transduced with PROX1, ATF5 and CYP3A4	Matrigel	48h/NA	APEL medium	Velazquez et al., 2021	
	hESCs & hiPSCs	Matrigel/ULA Plate	5–6d/>48d	DMEM/F12 medium, GlutaMAX, HEPES, N2, B27, BSA, N-Acetylcysteine, gastrin, Nicotinamide, CHIR99021, FSK, FGF10, HGF, A03-01, R-spondin1, EGF	Kim et al., 2022	
	Patient liver cancer cells	Matrigel	14–21d/NA	DMEM/F12, GlutaMAX, PS, N-Acetylcysteine, HEPES, EGF, FGF7, FGF10, HGF, Dex, Y27632	Sun et al., 2024	
Pancreas	Mouse adult duct cells	Matrigel	7d/9mo	AdDMEM/F12 medium, B27, N-Acetylcysteine, gastrin, EGF, RSPO1, Noggin, FGF10, Nicotinamide	Huch et al., 2013	
	Patient tumor cells	Matrigel	7–8d/NA	AdDMEM/F12 medium, HEPES, GlutaMAX, PS, Primocin, N-Acetylcysteine, Wnt3a, RSPO1, Noggin, EGF, gastrin, FGF10, Nicotinamide, A83- 01	Boj et al., 2015	
	Mouse islet EpCAM+cells	Matrigel	NA	DMEM/F12 medium, PS, B27, ITS, EGF, heparin, FGF2, VEGFa	Wang et al., 2020	
	hiPSCs	Matrigel	6-7d/30d	3D Kelco Gel Stem TeSR, FSK, dexamethasone, TGF- β RI kinaseinhibitor II/Alk5 inhibitor II, Nicotinamide, 3,3′,5-triiodo-l-thyronine sodium salt (T3), B27, R428, zinc sulfate, N-Cys	Yoshihara et al., 2020	
	hESCs & hiPSCs	Feeder cells	3–5d/>11d	DMEM, B27, EGF, bFGF, 616452, I-BET151	Ma et al., 2022	
	Patient pancreatic cancer cells	Matrigel	NA	DMEM/F12, HEPES, Glutamax, Primocin, A83–01, EGF, mNoggin, FGF10, Gastrin, N-Acetylcysteine, Nicotinamide, B27, R-spondin1 CM, Afamin/Wnt3A CM	Demyan et al., 2022	
Colon	mASCs	Matrigel	7–8d/NA	AdDMEM/F12 medium, Crypt culture medium, EGF, RSPO1, Noggin	Sato et al., 2009	
	hPSCs	Matrigel	13–14d/>140d	RPMI1640 media, dFBS-DMEM/F12, FGF4, Wnt3a, Activin A, RSPO1, Noggin, EGF, L-Glutamine, HEPES, N2, B27, PS	Spence et al., 2011	
	hiPSCs	Matrigel	NA/2mo	AdDMEM/F12 medium, GlutaMAX, HEPES, PS, B27, Y-27632, Streptomycin, Noggin, EGF, RSPO1, Wnt3a	Fordham et al., 2013	
	hPSCs	Hydrogel	10–14d/40d	AdDMEM/F12 medium, N2, B27, HEPES, L-Glutamine, PS, Noggin, SB431542, FGF2, Sant-2, SU5402, SHH, SAG	Cruz-Acuña et al., 2017	
	hPSCs	Matrigel & ULA plate	2–3d/6mo	DMEM/F12 medium, LDN193189, CHIR99021, EGF, B27, GlutaMAX, HEPES, PS	Han et al., 2021	
	Patient intestinal cells	Matrigel	NA	Wnt-3a or hAFM/Wnt-3a conditioned medium, RSPO1, Noggin, EGF, B27, N-Acetylcysteine, Nicotinamide, SB202190, A83-01, prostaglandin E2, gastrin, Primocin	Cho et al., 2021	
	Patient colon cancer cells	Matrigel	7–10d/NA	DMEM/F12, GlutaMAX, PS, N-Acetylcysteine, HEPES, B27, Noggin, EGF, A83-01, gastrin, SB202190	Parseh et al., 2022	
Kidney	Mouse embryonic kidney cells	Cell pellet	NA	KCM, MEM, FBS, PS, Y27632, Glycyl-H1152,	Unbekandt & Davies, 2010	
	mESCs & hiPSCs	ULA plate	2 d/18d	IMDM, F12 medium, N2, B27, PS, BSA, glutamine, ascorbic acid, 1-thioglycerol	Taguchi et al., 2014	
	hiPSCs	Matrigel	2–3d/3mo	APEL medium, CHIR99021, FGF9, bFGF, heparin	Takasato et al., 2015	
	hESCs	ULA plate	2-3d/43d	DMEM/F12 medium, CHIR99021, FGF9, Noggin. Dorsomorphin	Homan et al., 2019	
	Patient renal cell carcinomas cells	Matrigel	14–21d/NA	AdDMEM/F-12, Antibiotic-Antimycotic, GlutaMAX, HEPES, B-27, N-Acetylcysteine, Nicotinamide, SB202190, Y-27632, EGF	Li et al., 2022	
	Human fetal kidney cells	Matrigel	NA	Advanced DMEM, HEPES, Glutamax, Pen/Strep, B-27 supplement minus Vitamin A, N-acetylcysteine, R-spondin 1, EGF, A8301, CHIR99021, FGF10, GDNF, Heparin, LDN193189 dihydrochloride, Y27632	Gerli et al., 2024	
Brain	mESCs	ULA plate	NA/25d	DMEM/F12 medium, Neurobasal, pyruvate, 2-mercaptoethanol, Dkk-1, Lefty-1, Y27632, BMPRIA-Fc, SB431542, L-glutamine, B27	Eiraku et al., 2008	
	hPSCs	ULA plate	NA/10mo	DMEM/F12 medium, Neurobasal, N2, GlutaMAX, NEAA, Heparin, B27, 2-mercaptoethanol, insulin, retinoic acid	Lancaster et al., 2013	
	hPSCs	ULA plate	NA/>50d	DMEM/F12 medium, KSR, NEAA, GlutaMAX, 2-mercaptoethanol, PS, FGF2	Birey et al., 2017	
	hPSCs	ULA plate	2-3d/30d	Cortical differentiation medium, Glasgow-MEM, KSR, NEAA, pyruvate, 2-mercaptoethanol, PS	Velasco et al., 2019	
	hiPSCs	Matrigel	NA/>42d	DMEM/F12 medium, Neural basal medium, N2, B27, RA, GlutaMAX, MEM, Dorsomorphin, insulin, SB431542, PS, 2-mercaptoethanol	Gabriel et al., 2021	
	hESCs	ULA Plate	19d/>25d	DMEM/F12 medium, Neurobasal medium, PS, GlutaMAX, B27 without Vit A, EGF, FGF2	Pagliaro et al., 2023	
	Human fetal central nervous system (brain and spinal cord) cells	ULA Plate	14-21d/NA	AdDMEM/F12 medium, Neurobasal medium, PS, GlutaMAX, HEPES, B27 without VA, N2, MEM, FGF10, EGF, Primocin	Hendriks et al., 2024	
Retina	hESCs	ULA plate & Matrigel	NA/18d	NR culture medium, G-MEM, KSR, NEAA, pyruvate, 2-mercaptoethanol, PS, IWR1e, FBS, SAG, CHIR99021	Nakano et al., 2012	
	hPSCs	Matrigel	2d/24d	DMEM/F12 medium, B27 (without vitamin A), antibiotic-antimycotic, GlutaMAX, NEAA	Regent et al., 2020	
	hiPSCs	Geltrex	2d/>30d	DMEM/F12 medium, FBS, taurine, B27, NEAA, GlutaMAX, antibiotic-antimycotic, retinoic acid	Xie et al., 2020	
	hESCs	ULA plate	3d/>80d	GMEM, KSR, N2, ascorbic acid	Savoj et al., 2022	
	hiPSCs	Maxgel	8d/>120d	DMEM/F12, N2, NEAA, GlutaMAX, heparin, BMP4	Bohrer et al., 2023	
	hiPSCs	rhVTN-N	NA/>42d	E6 medium, N2	Gozlan et al., 2023	
Skin	mPSCs	ULA plate	NA/>56d	AdDMEM/F12 medium, N2, GlutaMAX, Normocin, LDN, FGF2, BMP4, SB431542	Lee et al., 2018	
	hPSCs	ULA plate	NA	AdDMEM/F12 medium, Neurobasal, E6 basal medium, GlutaMAX, B27, N2, 2-mercaptoethanol, Normocin, BMP4, SB431542, bFGF	Lee et al., 2020a	
	Human primary epidermal cells from foreskintissues	Matrigel	NA/42d	AdDMEM/F12 medium, BSA, B27, HEPES, GlutaMAX, N-Acetyl-Lcysteine,Forskolin, EGF, Wnt3a, A83-01	Wang et al., 2021b	
	miPSCs	Matrigel	2d/29d	AdDMEM/F12 medium, HEPES, Glutamax, B27, N-acetylcysteine-1, Noggin, Rspondin-1, FGF1, heparin, Forskolin	Kwak et al., 2024	
Bone	hPSCs	Matrigel	2–3d/NA	DMEM, FBS, L-glutamine, NEAA, SP, ITS-X, ascorbic acid, 2-mercaptoethanol, bFGF, TGF- β1, BMP2, GDF5, PS	Tam et al., 2021	
	PHOs	Matrigel	NA/>4mo	DMEM, Endothelial medium, FBS, GlutaMAX, ascorbic acid, BOM, MCSF, RANKL	Abraham et al., 2022	
	hiPSCs	ULA plate	14d/>62d	APEL2, PFHM II, FGF2, BMP4	Lamandé et al., 2023	
	BMSCs from rats	ULA plate	NA	Chondrogenic Induction, sodium pyruvate, TGF- β3, dexamethasone, ITS premix, ascorbic acid-2-phosphate	Shen et al., 2024	
Heart	hPSCs	Matrigel	NA/15d	DMEM, B27, GlutaMAX, L-ascorbic acid 2-phosphate, PS, glucose, palmitic acid	Mills et al., 2019	
	mESCs	Matrigel	NA/15d	DMEM/F12 medium, PS, KSR, sodium pyruvate, 2-mercaptoethanol, l-glutamine, progesterone, β-estradiol, insulin, transferrin, selenite, FGF4	Lee et al., 2020b	
	hPSCs	ULA plate	NA/21d	CGM media, B27, thrombin, cardiotrophin-1, EGF, FGF-2, Wnt-3A, EDN1	Ho et al., 2022	
	hiPSCs	gelatin	NA/25d	RPMI 1640, BSA, lascorbic acid 2-phosphate, sodium DL-lactate	Seguret et al., 2024	
Notes.

BME basement membrane extract

D dimensional

d day

ESC embryonic stem cell

h human

iPSCs induced pluripotent stem cells

KSR Knockout Serum Replacement

m mouse

mo month

NA not available

PHCs primary human cholangiocytes

PHHs primary human hepatocytes

PHOs primary human osteocytes

PMHs primary mouse hepatocytes

PS penicillin streptomycin

PSCs pluripotent stem cells (iPSCs &ESCs)

ULA ultra-low attachment

Given the significant challenges in optimizing conditions for organoid expansion, bioreactor-based suspension culture systems have recently attracted attention. These systems, which are widely used for large-scale cell culture in the pharmaceutical industry, offer potential solutions. The dynamic culture model of the bioreactor enhances the exchange of oxygen, nutrients, and metabolites. This improvement is believed to solve the limitations of static culture and markedly improve organoid survival and development (Schneeberger et al., 2020). More importantly, the relatively homogeneous spatial distribution of these components and signaling molecules greatly improves the homogeneity and reproducibility of the organoid products. Transferring static 3D culture to a dynamic spinner flask results in the growth of LGR5-positive liver stem cells achieved a dramatic 7-fold expansion enhancement. This new system also markedly upregulates functional maturation following differentiation (Schneeberger et al., 2020). The success of hPSCs suspension expansion over the past decade (Burrell et al., 2019) has inspired the possibility of extending the 3D culture system to include direct lineage differentiation. In this context, a 3D hPSC expansion system efficiently differentiated definitive endoderm (DE) cells—common intermediates in endoderm lineage differentiation—into 3D floating aggregates in a single batch using a bioreactor. This process yielded 1× 108 DE cells with over 92% purity, facilitating downstream production of various endodermal lineages, including hepatic, pancreatic, and intestinal lineages (Sahabian et al., 2021). Recently, a cascade 3D suspension system was developed that enables both the expansion and subsequent hepatic differentiation of hPSC-DEs. This system facilitated the massive production of hepatic organoids with high purity, over 85% and 93% for hepatocyte and cholangiocyte specification, respectively, in a 300 mL spinner flask (Feng et al., 2020). Although not specifically mentioned, this system may also permit the differentiation of other endodermal lineages under proper downstream differentiation conditions. Additionally, scaling up of cultures for various other lineages, including macrophages, pancreatic cells, and endothelial cells, is rapidly developing (Dossena et al., 2020; Gutbier et al., 2020; Takebe et al., 2017).

Nevertheless, it is worth noting that agitating bioreactors, which allow precise control of dissolved oxygen and nutrients, turbulence, and shear stress, have been rarely reported except in the context of iPSC and iPSC-platelet production (Suzuki et al., 2020). One possible reason is the limitations in refining the optimal bioreactor parameters for distinct iPSC-derived lineage progenitors. Additionally, the high cost and complexity of stage-specific medium supplements may further hinder scaling. A comprehensive understanding of the dynamic culture conditions required for organoid expansion and differentiation, along with integrated automated and programmed culture systems, is urgently needed to scale up organoid production. Evaluating organoid quality—including size, cellular components, and functionality—should be emphasized to ensure reproducibility between different passages and batches.

Automated and controllable culture system

With the development of organoid generation strategies (Fig. 1), the corresponding culture systems have seen significant improvements. However, the reproducibility and consistency of organoid culture systems remain major bottlenecks. Conventional culture processes involve a large number of human factors, low automation, poor organoid controllability, and human error, which lead to significant differences between different cell lines and organisms, resulting in numerous uncertainties and differences in organoid structure and function (Jiang et al., 2020). Automated culture may offer a solution to avoid variable errors caused by inconsistent manual handling during tedious experiments, allowing for precise mechanical automation (Park, Georgescu & Huh, 2019). For example, using an automated liquid-handling robot system, researchers established a fully automated and high-throughput screening-compatible platform with integrated imaging and analysis processes, which for the first time enabled the differentiation and formation of organoids (Czerniecki et al., 2018). Importantly, the Swiss Federal Institute of Technology in Lausanne has developed a high-throughput automated microcavity array technology. This technology is used for the high-throughput derivation of epithelial organoids within a polymer-hydrogel substrate. It clearly demonstrates that organoids generated by automated manipulation display reduced variability and increased time efficiency. Therefore, extending the scope of research relies on stable and reliable organoid culture and facilitates the scaling of culture systems (Brandenberg et al., 2020). Recently, a high-content screening (HCS) platform that allows researchers to screen drugs or other compounds against three-dimensional (3D) cell culture systems in a 384-well multi-well format has been established. This platform enabled automated, imaging-based HCS of 3D cellular models in a non-destructive manner, opening the path to complementary analysis through integrated downstream methods (Bozal et al., 2024).

Figure 1 Strategies to generate organoid systems.

Under specific 3D microenvironment (suspension culture, ECM, microparticles, etc.), tissue-derived stem/progenitor cells have been used for generating organoids with the aid of supportive cells. However, the limited cell source, along with challenges in scalability and reproducibility, presents significant hurdles. Pluripotent stem cells, including ESCs and iPSCs, offer a promising alternative due to their potential for efficient and scalable organoid production. PSCs could be synchronously differentiated towards three dermal layers, which facilitates the generation of more complex organoid structure with multiple cellular components. Following embryonic body formation, the organoid functionality and maturation could be significantly improved. With recent advancements in gene editing technologies, such as CRISPR/Cas9, increasing types of iPSCs-derived diseased organoids have been established by means of gene mutation/ knock-in or out. This progress greatly extends the applications in personalized disease modeling and drug testing. In addition, for ESC or iPSC-derived organoids, HLA knockout is expected to create universal organoids, which are expected to be useful in organ transplantation. ECM, extracellular matrix; ESCs, embryonic stem cells; HLA, human leukocyte antigen; iPSCs, induced pluripotent stem cells; KI, knock in; KO, knock out; PSCs, pluripotent stem cells, included iPSCs and ESCs; TF, transcription factors.

To achieve precise and automatic control over organoid generation, culture, and analysis conditions, an organoid-on-chip system has been developed, which is based on a microfluidic cell culture device manufactured using a microchip fabrication method. This system consists of multiple microchambers with fluid flow, a variety of living cells, mechanical force stimulation, and other complex factors in vitro. It simulates the main structural and functional characteristics of human organs] by implanting human cells into microfluidic chips for 3D culture (Shirure, Hughes & George, 2021). However, microenvironments such as blood vessels, stromal components, immune cells, and neuroendocrine microenvironments are commonly lacking and need to be further complemented to create more realistic conditions (Schuster et al., 2020). The Tsinghua-Berkeley Shenzhen Institute (TBSI) has combined organoid generation with an organ-on-a-chip system and 3D printing technologies to open up a new field for organoid scaling. In this study, a microfluidic droplet system was used to shear cell-containing Matrigel into homogeneous microspheres, which were then followed by 3D printing. This enabled the rapid generation, culture, and automated manipulation of organoids, leading to the development of homogeneous, controlled, high-throughput, and scalable tumor organoid systems. Nevertheless, this high-throughput and automated organoid culture system still has some limitations, such as a relatively low success rate of organoid printing, incomplete automation, and the need for additional manual transfer (Jiang et al., 2020). Recently, a systematic approach has been taken to investigate the initial seeding density of endothelial cells and its effects on interconnected networks, which has been combined with hepatic spheroids to develop a liver-on-a-chip model. This system provides insight into potential hepatotoxicity caused by various drugs and allows for the assessment of vascular dysfunction in a high-throughput manner (Wang, Andrade & Smith, 2023). In addition, since the organoid microarray model is constructed in a predefined manner, its ability to capture the dynamics of organoid development in response to drugs or environmental changes is very limited. Integrated engineering techniques are needed to monitor and analyze the dynamic development in micro-engineered organoid cultures. Furthermore, due to the complexity of multicellular nutrient and signal requirements in organoids, future optimization of media composition and stepwise culture conditions are necessary for the co-culture system. On this basis, the establishment of standardized good manufacturing practice (GMP) guidelines is expected to further ensure the quality and biosafety of organoids, eventually driving the shift from laboratory research to clinical or industrial applications.

Frontiers and perspectives

Organoid atlas

Organoids have tremendous potential for biomedical research due to their unique advantage of organ-like three-dimensional structures to mimic the physiology and function of native organs (Ashok et al., 2020; Lancaster & Knoblich, 2014). To fully realize their potential and address practical application challenges, organoids should be characterized and validated as reliable organ substitutes (Bock et al., 2021). However, current organoid techniques still face many limitations, including inadequate multilineage development, incomplete neuroendocrine and immune systems, undeveloped vascularization networks, suboptimal physiological functions, and drug responses. A suite of recently developed integration of single-cell and spatial profiling may offer new prospects for addressing these existing problems. These methods allow for identifying the spatial distribution of multiple cell populations within tissues or organoids, and capturing individual cell physiological characteristics and intercellular communication networks by physically localizing cell transcriptomic information within specific spatial locations (Longo et al., 2021). On this basis, comprehensive spatial mapping of liver tissue has identified transcriptome-wide zonation of parenchymal and non-parenchymal liver cells, including previously unknown subpopulations (Aizarani et al., 2019). Moreover, in-depth bioinformatic analysis revealed that the heterogeneous EPCAM+ liver population includes hepatocyte- and cholangiocyte-biased cells, as well as a unique TROP2int population that exhibits bipotency and organoid formation capability, potentially providing a novel liver progenitor source for generating organoids. Notably, the atlas also revealed the differences in gene expression and function between normal and hepatocellular carcinoma livers across multiple cell types, providing new insights into inter-cellular communications during liver cancer development and aiding in the design of in vitro modeling (Aizarani et al., 2019). Additionally, single-cell sequencing and spatial transcriptomics are increasingly used to characterize tissue architecture and biological mechanisms in various normal and cancerous tissue/organs (Mutuku et al., 2022; Rao et al., 2021; Yamada & Nomura, 2020) and to deconvolute the microenvironment for tissue homeostasis and cancer progression (Ayyaz et al., 2019; Baccin et al., 2020; Wang et al., 2021a).

On this basis, comparative spatial molecular profiling of organoids with their corresponding tissues or organs may fundamentally provide references for the revolutionary outbreak of current protocols and strategies for organoid generation, culture, and even disease modeling. For example, this approach can identify the specific roles of individual cell populations and reveal key regulatory mechanisms in organogenesis or carcinogenesis. Indeed, many previously unobservable discrepancies in gene expression and organization of particular cellular composition, have been confirmed between cultured organoids and relevant organs by in-depth transcriptomic comparison at the single-cell level. These findings offer significant clues for improving organoid models through advanced engineering and culture systems (Kanton et al., 2019) Moreover, the organoid atlas provides a scalable tool for evaluating organoids in terms of constructure, functionality, and even unexpected mutations, which is crucial for quality control, particularly in terms of regenerative therapy (Sridhar et al., 2020). To create comprehensive reference maps for all human cells as a basis for understanding human biology and treating diseases, an Organoid Cell Atlas pilot project was recently launched in Europe. This project is expected to complement the profiling of primary tissues with perdurable models for studies in biomedical discovery and regenerative therapies (Bock et al., 2021). Although spatially resolved transcriptomic technologies are now being widely adopted, their application is still facing long-term challenges. Current approaches cannot provide deep transcriptomic information on precisely localized single cells in tissues or organoids (Rao et al., 2021).

Moreover, there is an urgent need to develop novel computational methods for analyzing spatially resolved transcriptomic data. These methods should characterize the heterogeneity of cells within their spatial contexts and to derive biological insights into organoids and their derivation models, as well as the corresponding primary tissues (Longo et al., 2021).

Universally compatible iPSC-organoids and biobank

Organoids can be established from autologous iPSCs, which are immunologically identical to the donor, offering significant immunological advantages in transplantation therapy. However, creating autologous iPSC from each patient’s somatic cells is difficult for standardized treatments due to low efficiency and high cost (Murata et al., 2020). With the development of gene editing tools, the possibility of iPSCs transplantation without allogeneic rejection has become a key focus in regenerative medicine (Ichise et al., 2017; Koga, Wang & Kaneko, 2020; Morizane et al., 2017; Zhao et al., 2011). In particular, CRISPR/Cas9 gene editing tool enables the creation of universally compatible iPSCs by remodulating immune-related antigens via gRNA targeting and cas9 nuclease-mediated shearing (Chen et al., 2019). Human leukocyte antigen (HLA), which plays an important role in distinguishing self from non-self, is a major barrier in organ and cell-based transplantation. It is commonly believed that HLA mismatch between the donor and recipient is a major barrier to organ or cell-based transplantation, while matched HLA can significantly reduce the risk of graft rejection and graft-versus-host disease (GVHD), and improve allograft survival (Koga, Wang & Kaneko, 2020). HLA class I and II complexes mediate antigen-specific adaptive immune responses and act as ligands for T and NK cells, which differentiate between self and non-self components (Long et al., 2013). To overcome immune rejection, β2-microglobulin (B2M), a common protein subunit, essential for HLA class I expression, is commonly knocked out using traditional editing. However, knocking out B2M silences of all HLA class I molecules, causing “miss self” and leading to NK cell attack (Flahou et al., 2021). To address this, immune rejection was suppressed by disrupting the HLA-A and -B alleles and HLA class II molecules, while retaining HLA-C, which helps avoids NK cell-mediated self-attack. This approach enables iPSCs to evade T and NK cell attacks in vitro and in vivo (Lee et al., 2020a). iPSCs produced by the iPS Cell Research and Application Center of Kyoto University using a gene knockout strategy with CRISPR/Cas9 exhibited extremely low immunogenicity and could evade attack in vitro and in vivo after T and NK cells differentiate into platelets (Xu et al., 2019). Similarly, another group used CRISPR-Cas9 to knock out beta-2-microglobulin (B2M) in kidney organoids, successfully protecting kidney organoids derived from these iPSCs against T-cell rejection (Gaykema et al., 2024). These successes highlight the possibility of using universally compatible iPSC sources to generate highly biocompatible and available organoid derivatives to solve the current extreme shortage of organs for transplantation, with dramatically reduced GVHD.

However, challenges remain in current HLA knockout strategies. Precise targeting of HLA-A and HLA-B while retaining HLA-C is crucial to avoid off-target effects on HLA-C homologous sequences. In addition, the potential reduction in cell viability, proliferation, and pluripotency of PSCs should be monitored and controlled to maintain the full differentiation potential of PSCs. Moreover, the general problems of low transfection efficiency and off-target phenomena in gene editing must be optimized (Chen et al., 2019). To enhance on-target specificity, scientists have attempted to modify the Cas9 protein to alter PAM preferences or enhance target DNA recognition and developed systems to regulate Cas9 expression during transcription and translation (Kleinstiver et al., 2016; Kleinstiver et al., 2015; Shen et al., 2018). Research on trophoblast organoids has advanced the understanding of placental development. Organoid models using CRISPR/Cas9 technology examined the role of HLA-G in trophoblast function and differentiation. JEG-3 trophoblast organoids (JEG-3-ORGs) were established, expressing key trophoblast representative markers and had the capacity to differentiate into EVT. HLA-G knockout (KO) via CRISPR/Cas9 significantly altered the trophoblast immunomodulatory effect on NK cell cytotoxicity, as well as the trophoblast regulatory effect on HUVEC angiogenesis (Zhuang et al., 2023). However, the current technology is still immature, and future improvements in specificity, targeting efficiency, and use of highly efficient, biocompatible, and non-immunogenic delivery vehicles are needed. Considering the overall usefulness of the CRISPR-Cas9 gene editing tool in terms of efficiency and biocompatibility, its use in in vivo transplantation is believed to enhance safety and efficacy (Chen et al., 2019).

Given the rapid demand for organ/tissue models and transplantation grafts for diverse research and clinical applications, the establishment of organoid biobanks offers extensive opportunities for ready-to-use tools and sources (Perrone & Zilbauer, 2021). Large-scale production of universally compatible iPSC-organoid is expected to expand transplantation therapies to many patients with different HLA backgrounds (Fig. 2). However, due to HLA’s high polymorphic and variation among different ethnic groups in different regions, creating organoid banks with sufficient HLA haplotypes that match a wide range of populations remains challenging (Flahou et al., 2021). Recruiting HLA-homozygous donors to cover diverse populations is particularly difficult. Biobanks must consider rare frequency alleles and include rare donors, with each cell source carefully characterized and evaluated for regulatory safety (Flahou et al., 2021). Recently, a PLC biobank with 399 tumor organoids derived from 144 patients was established, which recapitulates histopathology and genomic landscape of parental tumors, and is reliable for drug sensitivity screening. This study explored PLC heterogeneity, developed predictive biomarker panels, and identified a lenvatinib-resistant mechanism for combination therapy (Yang et al., 2024). Beyond identifying clinical biocompatibility and safety, considerable work remains to establish standard, verified clinical-grade allogeneic organoid biobanks. Global collaborative efforts from the scientific, clinical, and industrial communities are required to accelerate the development of this encouraging field.

Figure 2 Universally compatible iPSC-organoid biobanking and applications.

iPSC could be reprogrammed from a patient’s somatic cells and used as a starting source for producing patient-derived multiple desired organoids. Advanced 3D organoid culture system contains multiple supportive cell populations such as stromal cells, endothelial cells, as well as the neuroendocrine and immune system, allowing for a closer approximation to in vivo organs. The inclusion of multiple supportive cell populations contributes to the construction of disease models and the development of pharmaceutical products at this stage. The automated culture system provides the possibility to overcome the variable errors caused by manual inconsistency, and scale-up of support organoid production, and is expected to provide a solution to the difficult breakthrough of large-scale expansion of standardized organoids at the GMP level. In addition, the use of gene-editing tools to knock out immune response antigens such as HLA is expected to generate universally compatible iPSC-organoids ideal for allogeneic transplantation. Combining single-cell and spatial profiling, organoid mapping can provide structural and molecular profiles of organoids in comparison with corresponding tissues or organs. This approach enhances the high simulation of current organoid construction and cultivation. Additionally, it contributes to further optimization of disease modeling. The establishment of organoid libraries will greatly contribute to the provision of ready-to-use disease models for drug screening. These libraries are also expected to provide immediate organoid substitutes for the treatment of malignant or advanced diseases, such as cancer. The establishment of organoid banks will greatly help to supply ready-to-use disease models for drugs screening, and is expected to provide immediate organ substitutes for treating malignant or late-stage diseases such as cancer.

Interdisciplinary collaboration

The future of organoid research is deeply intertwined with interdisciplinary collaboration, which integrates expertise from diverse fields to overcome current limitations in scalability, functionality, and clinical applications. While the biological sciences provide the foundation for understanding the molecular and cellular mechanisms of organoid formation, other disciplines, such as bioengineering, materials science, and computational biology, play essential roles in advancing this technology.

Bioengineering and material science.

One of the most promising areas of collaboration is between bioengineers and materials scientists, particularly in the development of synthetic scaffolds and extracellular matrices. These artificial matrices mimic the natural environment of human tissues, promoting the growth, differentiation, and organization of cells within organoids. For example, bioengineers are utilizing 3D bioprinting technologies to construct scaffolds that provide mechanical support and guide the spatial organization of organoid structures (Deng et al., 2024). These innovations are critical for scaling up organoid cultures for high-throughput drug screening and clinical-grade tissue production.

Material scientists contribute by designing tunable hydrogels with adjustable stiffness, porosity, and biochemical signals to simulate the tissue-specific extracellular matrix, improving organoid maturation and functionality (Nerger et al., 2024). These advancements enable the generation of more complex organoid models that better mimic the in vivo environment, an essential step for applications in regenerative medicine and disease modeling.

Computational biology and systems medicine.

The advent of big data and artificial intelligence is revolutionizing organoid data analysis, making collaboration with computational biologists increasingly vital. Organoids generate vast datasets from high-throughput genomics, proteomics, and transcriptomics studies. Computational biologists are developing advanced algorithms and machine learning models to analyze these data, identifying key signaling pathways in organoid development and disease progression (Santamaria et al., 2023; Wahle et al., 2023). This collaboration supports precision modeling, allowing organoids to be tailored to reflect patient-specific genetics or disease conditions, a powerful approach in personalized medicine.

In addition, systems biology approaches are integrating multi-omics data from organoid studies to map cellular networks and predict drug responses. For instance, modeling the interactions between different cell types within an organoid (e.g., immune cells in liver or brain organoids) is crucial for understanding diseases like cancer and neurodegeneration. These insights may reveal new therapeutic targets that traditional models cannot capture.

Ethical and regulatory concerns

Use of human stem cells.

Organoids are often derived from induced pluripotent stem cells (iPSCs) or embryonic stem cells (ESCs), raising ethical concerns about the use of human embryos or reprogramming adult cells. Regulatory frameworks governing stem cell research vary across countries, with some enforcing stricter guidelines on the sourcing and manipulation of human cells (Park et al., 2024). For instance, research involving ESCs may conflict with certain cultural or religious beliefs, necessitating transparent and culturally sensitive regulations to address these concerns.

Clinical applications.

While organoids hold significant potential for regenerative medicine, concerns about safety, efficacy, and the long-term impacts of transplanting lab-grown tissues into humans remain. Regulatory bodies like the FDA and EMA have yet to fully establish guidelines specific to organoid-based therapies, complicating the path to clinical application. Navigating these regulatory challenges requires strict quality control measures, ethical sourcing of stem cells, and comprehensive preclinical testing.

Consent and data privacy.

The use of iPSCs involves obtaining consent from donors, as their genetic information is encoded within the cells. Ensuring informed consent, especially regarding the future uses of organoids derived from their cells, while also protecting donor privacy in light of genomic data sharing is critical regulatory concern. This issue is particularly significant in precision medicine, where organoids might be used to model individual patients’ conditions.

To address these ethical and regulatory challenges, interdisciplinary collaboration is essential not only within scientific fields but also with ethicists, legal experts, and policymakers. Regulatory frameworks must be updated to keep pace with advancements in organoid technology, ensuring that innovation remains both ethically and scientifically sound. This might involve creating new standards for informed consent, safety regulations for organoid-based therapies, and defining ethical boundaries for the extent of human tissue mimicry.

Conclusions

As organoid research progresses, ensuring reliability, efficiency, and scalability has become the focus of organoid research and applications. Although challenges remain, iPSC-derived multicellular organoids hold promise for drug screening and transplantation therapy. The integration of bioreactors with automated culture systems may greatly increase the scale of organoid production. Additionally, it is expected that future advances in organoid atlas and organoid biobanking may contribute to more reliable and practical applications in drug development and regenerative medicine.

We appreciate Drs. Li-Ping Liu, Yu-Mei Li, Di Cao, Mei Fang and Hang Zhou in Jiangsu University, Mr. Shang-Ping Tian, Mr. Yu-Mu Song and Ms Ji-Yue Yan in Wuyi University for contributing research assistance and discussion.

Additional Information and Declarations

Competing Interests

Author Contributions

Data Availability

The authors declare there are no competing interests.

Jian-Yun Ge performed the experiments, analyzed the data, prepared figures and/or tables, and approved the final draft.

Yun Wang performed the experiments, analyzed the data, prepared figures and/or tables, and approved the final draft.

Qi-Lin Li analyzed the data, prepared figures and/or tables, and approved the final draft.

Fan-Kai Liu analyzed the data, prepared figures and/or tables, and approved the final draft.

Quan-Kai Lei analyzed the data, authored or reviewed drafts of the article, and approved the final draft.

Yun-Wen Zheng conceived and designed the experiments, authored or reviewed drafts of the article, and approved the final draft.

The following information was supplied regarding data availability:

This is a literature review.

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
