# Peer review of "Trends and challenges in organoid modeling and expansion with pluripotent stem cells and somatic tissue"

_PeerJ, doi:10.7717/peerj.18422_

## Round 0.1 · original submission · Major Revisions

· Academic Editor

Major Revisions

The authors are requested to carefully revise the manuscript and answer the questions raised by the reviewers.

Reviewer 1 ·

Basic reporting

This manuscript provides an extensive overview of the progress in organoid generation, their applications in basic and preclinical research, automated culture techniques, and universally compatible organoid biobanks. Before publication, the following improvements are suggested.

1. Rewrite and simplify the Abstract section. There are some repetitive contents, and the phrase 'Based on previous experiences in our lab' should be removed, as most of the cited literature is from other research groups
2. Add the latest references in: (a) New techniques and methods in organoid modeling and expansion. (b) The latest clinical applications or preclinical studies, particularly those that have advanced to clinical trials. (c) Emerging applications of organoid technology in fields such as personalized medicine and toxicology research.
3. Expand the Discussion on applications, including toxicology research, drug metabolism and pharmacokinetics (DMPK) studies, and rare disease research.
4. Expand the Discuss on the Prospects of Interdisciplinary Collaboration
5. Add a section addressing ethical and regulatory considerations.

Experimental design

no comment

Validity of the findings

no comment

Reviewer 2 ·

Basic reporting

This review manuscript summarizes Trends and challenges in organoid modeling and expansion with pluripotent stem cells and somatic tissue. The review has discussed important achievements, challenges and limitations, and future perspectives in organoid modeling with extensive literature reviewing and detailed table and figure illustrations.
The manuscript may benefit from addressing a major issue, that is the overall structure of the review.
(1) In both abstract and main text, there are confusions in putting “achievements and limitations” and “perspectives and challenges” in the same sections.
(2) It may make more sense to lay out the article with the main structure of “advancement/progress”, “limitations/challenges”, and “future perspectives”.
(3) The title of every subsection under the three main themes may also benefit from adding extra description to clarify what those subtitles mean. For example, “Multicellular organization and extracellular matrix” could be changed to “Progress in integrating multicellular organization and extracellular matrix in organoid modeling” Or “challenges in …”
(4) The abstract and figures may follow the same structure to emphasize the three major themes.

Experimental design

No comment

Validity of the findings

No comment

Reviewer 3 ·

Basic reporting

In this review paper, Jian-Yun Ge and co-authors summarize recent advancements in the development of organoid culture systems. They focused on the methods of organoid
Generation, expansion of organoids for disease modeling and therapy. Overall, this kind of review is timely for the readers to get familiar to the literatures in organoid systems and their applications in basic, translational and pharmaceutical research. The tables and figures provide useful illustration and concise summary. I believe that after extensive revision, this review would be publishable in this journal.

Experimental design

My criticism for this manuscript is two-fold.
First, the authors may streamline the organization of this manuscript with more subtitles. This allows a smooth flow of the topics to be discussed and reviewed. The authors may comment on the pros and cons in the summary tables of organoid cultures and highlights the optimal methods in their opinions.

The second concern is English language. The authors shall consult with a native English editor to short their sentences for the easy reading. The language can be polished extensively across the entire manuscript. Standard terminology should be used instead of atypical wording.

Validity of the findings

Fine

Additional comments

Extensive English editing is required.

Below are just two examples:


1. Incomplete sentence in Line 113. such as “Literature with publication date in the last decade was
more.”

2. Atypical wording in Line128 “…devoid of genome-stability impairment” should be simplified as “Genome instability”

---

## Round 0.2 · accepted · Accept

· Academic Editor

Accept

After revisions, one reviewer agreed to publish the manuscript. I also reviewed the manuscript and found no obvious risks to publication. Therefore, I also approved the publication of this manuscript.

Reviewer 3 ·

Basic reporting

This revision has addressed most of concerns from this reviewer. I recommend it to be accepted for publication now.

Experimental design

The format of this review paper is fine.

Validity of the findings

Fine